# The Importance of Optical Coherence Tomography in the Diagnosis of Atypical or Subclinical Optic Neuritis: A Case Series Study

**DOI:** 10.3390/jcm12041309

**Published:** 2023-02-07

**Authors:** Yumin Huang-Link, Ge Yang, Greta Gustafsson, Helena Gauffin, Anne-Marie Landtblom, Pierfrancesco Mirabelli, Hans Link

**Affiliations:** 1Division of Neurology, Department of Biomedical and Clinical Sciences, Linköping University, 581 85 Linköping, Sweden; 2State Key Laboratory of Ophthalmology, Zhongshan Ophthalmic Center, Sun Yat-sen University, Guangzhou 510275, China; 3Division of Neurophysiology, Department of Biomedical and Clinical Sciences, Linköping University, 581 85 Linköping, Sweden; 4Division of Neurology, Department of Medical Sciences, Uppsala University, 752 36 Uppsala, Sweden; 5Division of Ophthalmology, Department of Biomedical and Clinical Sciences, Linköping University, 581 85 Linköping, Sweden; 6Department of Neurosciences, Karolinska Institutet, 171 77 Stockholm, Sweden

**Keywords:** subclinical optic neuritis, optical coherence tomography, peripapillary retinal nerve fiber layer, macular ganglion cell-inner plexiform layer

## Abstract

**Background:** Optic neuritis (ON) is an inflammatory condition of the optic nerve. ON is associated with development of demyelinating diseases of the central nervous system (CNS). CNS lesions visualized by magnetic resonance imaging (MRI) and the finding of oligoclonal IgG bands (OB) in the cerebrospinal fluid (CSF) are used to stratify the risk of MS after a “first” episode of ON. However, the diagnosis of ON in absence of typical clinical manifestations can be challenging. **Methods and Materials:** Here we present three cases with changes in the optic nerve and ganglion cell layer in the retina over the disease course. (1) A 34-year-old female with a history of migraine and hypertension had suspect amaurosis fugax (transient vision loss) in the right eye. This patient developed MS four years later. Optical coherence tomography (OCT) showed dynamic changes of the thickness of peripapillary retinal nerve fiber layer (RNFL) and macular ganglion cell-inner plexiform layer (GCIPL) over time. (2) A 29-year-old male with spastic hemiparesis and lesions in the spinal cord and brainstem. Six years later he showed bilateral subclinical ON identified using OCT, visual evoked potentials (VEP) and MRI. The patient fulfilled diagnosis criteria of seronegative neuromyelitis optica (NMO). (3) A 23-year-old female with overweight and headache had bilateral optic disc swelling. With OCT and lumbar puncture, idiopathic intracranial hypertension (IIH) was excluded. Further investigation showed positive antibody for myelin oligodendrocyte glycoprotein (MOG). **Conclusions:** These three cases illustrate the importance of using OCT to facilitate quick, objective and accurate diagnosis of atypical or subclinical ON, and thus proper therapy.

## 1. Introduction

The intraocular portion of the optic nerve is unmyelinated before passing lamina cribrosa and represents axons of the central nervous system (CNS). This 1 mm thick layer of the axons in the optic disc head together with their neurons in the macula can be accurately measured in vivo with optical coherence tomography (OCT) both qualitatively and quantitatively [1,2]. OCT has been increasingly used to diagnose and monitor neurodemyelination such as multiple sclerosis (MS) [3] and neuromyelitis optica spectrum diseases (NMOSD) [4,5], in which involvement of the optic nerve is common [6]. Optic neuritis (ON) is an inflammatory process that occurs typically with acute or subacute monocular vision loss in young Caucasian females. Ocular or orbital pain is the most common associated symptom and is usually exacerbated by eye movement. Dyschromatopsia with red desaturation is a relatively common symptom of ON [7,8,9]. ON presents as a first event of MS in about 25% of patients and occurs during the relapsing-remitting phase in up to 70% of MS patients [10]. Evidence from postmortem studies showed that optic nerve involvement in MS is present in up to 99% of the cases [11]. Silent optic nerve atrophy and lesion have been identified in non-ON eyes in MS [10,12] by using imaging modalities indicating an omitted diagnosis of atypical or subclinical ON. Atypical ON at onset or subclinical ON during disease course is a challenge in diagnosis and lack of consensus, which may delay time to effective therapy [13]. In a recent position paper, OCT was proposed to be one of the three obligatory paraclinical methods to assist diagnosis of ON [13]. OCT as a high-resolution retinal imaging technique can quantify ten retinal layers including the retinal nerve fiber layer (RNFL) and the macular ganglion cell-inner plexiform layer (GCIPL) at the level of a few micrometers. RNFL and GCIPL represent the first-order neuron of the visual pathway, thus providing a natural window to evaluate the optic nerve in vivo [3,14,15]. In this article, we present three cases: two with clinically atypical ON and one with subclinical ON, demonstrating longitudinal changes of the peripapillary RNFL and GCIPL as measured with OCT in parallel with findings from ophthalmological examinations and neuroimaging to emphasize the expanding role of OCT in differential diagnosis and monitoring disease course.

## 2. Methods and Materials

Case 1. A 34-year-old obese Caucasian female with a history of migraine, hypertension and a family history of ischemic heart disease had bilateral headaches for 7 days. On the last day, she experienced painless blurry vision in the right eye. She presented no pain in the eye or on eye movement. Ophthalmoscope examination revealed normal eye bottoms. She was referred to neurologists due to suspect amaurosis fugax (transient vision loss) or migraine visual phenomena. Visual acuity was 0.1 in the right and 1.0 in the left eye. Color vision was normal. Visual field (VF) test using Humphery perimetry showed superior quadrantanopia in the right eye with a visual field index (VFI) of 74% (Figure 1a), which was further reduced to 54% within one week (Figure 1b) and was normalized spontaneously at two months’ follow-up (Figure 1c). MRI of the brain showed a T2 FLAIR lesion in the occipital lobe (Figure 1d). Cerebrospinal fluid (CSF) showed normal cell count, lactate, glucose and IgG index, but four specific oligoclonal IgG bands (OB) in CSF without correspondence in the serum. Optical coherence tomography (OCT) showed dynamic changes in the thickness of peripapillary retinal nerve fiber layer (RNFL) and macular ganglion cell-inner plexiform layer (GCIPL) over time (Figure 1f,g). One week after the onset of blurry vision, OCT showed normal thicknesses of RNFL of both eyes, but asymmetry with predominance in the right eye, which was similar at the 2-month follow-up. However, at the 5-month follow-up, thinning of RNFL thickness was observed in the right (from 101 to 91 µm), not in the left eye (93 vs. 94 µm). Similar trends of GCIPL thinning were also observed in the right eye (87 to 83 µm), and no thinning of GCIPL in the left eye (88 vs. 89 µm) (Figure 1h,i). Visual evoked potentials (VEP) showed delayed latency of P100 in the right eye one week after the onset (right 111, left 101 ms) (Figure 1j,k), which remained unchanged 5 years later (right 110, left 103 ms) (not shown). A diagnosis of atypical optic neuritis (ON) of the right eye was confirmed based on the findings from VF, OCT and VEP, thus establishing the diagnosis of clinically isolated syndrome (CIS) [10]. Four years later, the patient had a second relapse with numbness and tingling in the legs. Brain MRI showed 10 supratentorial periventricular T2 FLAIR lesions including a juxtacortical lesion (Figure 1e). The MS diagnosis was then fulfilled [16] and therapy with rituximab 500 mg twice per year was initiated. The patient had no relapses nor new changes on MRI or OCT during the 5-year follow-up.

Case 2. A 29-year-old Caucasian man was admitted to our neurological department with weakness in the left arm and leg. The symptoms developed successively over the last several days without obvious triggering factors. His previous medical and family history was not remarkable. Neurological examination revealed moderate spasticity in muscles of the biceps brachii and quadriceps femoris. Strength examination showed mild paresis in the arm and leg with dominance distally in the hand and foot. Ankle clonus, Hoffman’s and Babinski’s signs were positive on the left side. The higher cerebral functions, cranial nerves, sensory and cerebellar functions were intact. Blood tests were unremarkable. CSF twice during the first two weeks after onset showed slightly elevated albumin (355 respective 415 mg/L, reference < 320), normal cell count, lactate, glucose and IgG index, and no IgG specific OB were detected in the CSF. Brain MRI was normal, and spinal cord MRI showed hyperintensity with enhancement extending in C2–3 level (Figure 1a) lateral to the left side (Figure 2b). The patient received intravenous methylprednisolone 1 g/day for three days and oral baclofen 20 mg twice daily. Two months later, the patient had increased spasticity and weakness in the left extremities. Brain and spinal cord MRI showed new lesions with enhancement in the ventral medulla oblongata (Figure 2c). Repeated CSF and blood tests before therapy and during the relapses were not remarkable, including negative antibodies of NMOSD for aquaporin-4 (AQP4) IgG and myelin-oligodendrocyte glycoprotein (MOG) IgG. Analysis for human full-length MOG and AQP4 autoantibodies was performed using a commercial cell-based assay (CBA). Based on the suspect antibody-negative NMOSD [17,18], this patient was again treated with high-dose intravenous methylprednisolone 1 g daily for three days, followed by rituximab twice per year. He had no relapses, neither clinically nor on MRI, for five years. Due to recurrent respiratory tract and skin infections (erysipelas), rituximab was paused. One and half years later without rituximab, the patient experienced increased weakness in the left leg. MRI revealed new lesions with enhancement in C2–3 dorsolateral to the left side (Figure 2d) and in the optic chiasm (Figure 2e). Examination with OCT (Figure 2f) and VEP (Figure 2g) showed normal findings at 12 months’ follow-up after onset, but abnormal at the 18 months’ follow-up after the pause of rituximab (Figure 2h,i). The patient experienced no clinical symptoms of ON. His visual acuity was preserved bilaterally and color vision was slightly affected, and the VF test showed inferior bitemporal quadrantanopia (Figure 2j,k). Thus, subclinical bilateral ON was confirmed and diagnosis of seronegative NMOSD was fulfilled [19].

Case 3. A 23-year-old obese (body mass index 31) female of Caucasian descent was referred by ophthalmologist to our department due to five days’ history of headaches in the forehead and blurry vision in both eyes. She had no eye-movement-related pain. Color vision was well preserved. Ophthalmoscopy revealed bilateral optic disc swelling (Figure 3a,b). MRI of the brain was normal. Lumbar puncture was performed based on suspected idiopathic intracranial hypertension (IIH). CSF opening pressure was normal (15 cm water). Further MRI of the spinal cord was normal, but the orbital MRI visualized bilateral pathologic enhancement in the intraorbital optic nerves (Figure 3c). Blood and CSF tests were normal, except positive myelin oligodendrocyte glycoprotein (MOG) antibody. The OCT thickness map showed significantly thickened RNFL in both the right (Figure 3d) and left eye (Figure 3e) at day 7 that was thinned at week 6 and stable at follow-up at 3. 5 months and 2 years after onset. At day 7, GCIPL thickness was normal in both eyes at day 7 (Figure 3f,g), but was thinned at week 6 and remained unchanged at the follow-ups. Perimetry demonstrated peripheral lower VF defect in the right (VFI 88%) (Figure 3h) and normal (Figure 3i) in the left eye (VFI 99%). VF was mostly affected in both eyes at day 14 (VFI right 26%, left 21%) but recovered at week 6 (VFI right 98%, left 94%) when disc swelling vanished as shown with OCT (Figure 3d,e). VEP showed normal latency of bilateral P100 at one year’s follow-up). Thus, bilateral subclinical ON was confirmed with MRI, OCT and VF [10,20] and MOG-associated disease diagnosis was fulfilled according to ON classification proposed by Petzold et al. [13]. The patient was treated first with plasma exchanges five times, then with methylprednisolone 1 g/day for three days followed by rituximab twice per year. She had no relapse during three years’ follow-up.

## 3. Discussion

Case 1 presents atypical ON without eye pain nor dyschromatopsia, though there are risk factors for MS-associated ON (MS-ON) including age, sex and ethnic Caucasian background [8,13,21]. Hypertension and migraine together with a family history of heart disease mislead the diagnosis to amaurosis fugax/migraine with aura. This patient presented retrobulbar ON with normal optic disc, but VF showed monocular visional defect development during the acute phase of MS-ON, which recovered at week 8, although delayed P100 latency remained on VEP test. Thicknesses of RNFL and GCIPL in both eyes were normal but asymmetric. Follow-up showed dynamic changes of right RNFL with first thickening and then thinning, indicating optic nerve swelling at the acute phase followed by chronic atrophy. It took five months for RNFL thickness to become stable after ON. GCIPL showed symmetry in both eyes at onset and no swelling during the course. Thinning of right GCIPL was observed at week 8 and remained stable. Investigation for MS was motivated and completed within a short time, and CIS diagnosis was established. As expected [10,22], this patient converted from CIS into MS four years later and immune-suppressive therapy was initiated. This case indicates that dynamic changes of RNFL and GCIPL together with VF are strongly suggesting acute ON, even in absence of typical symptoms. Asymmetric RNFL and GCIPL on OCT together with delayed P100 latency on VEP are of help to diagnose previous ON [23], even when VF and color vision are normal. VF defects and dyschromatopsia are relatively common in acute ON and recovery is high. Abnormal VF and color vision are not helpful for the diagnosis of subclinical or chronic ON. Thinned RNFL and GCIPL visualized using OCT remain in general and are therefore valuable biomarkers [24]. It is helpful to combine structural OCT and functional VEP to diagnose and monitor suspect ON [25,26], but delayed P100 is more sensitive in acute ON due to optic disc swelling and often improves or recovers when disc swelling vanishes.

Case 2 had bilateral subclinical ON without visual symptoms for 6 years. During the first months, the patient had recurrent myelitis and enhancement lesions in the spinal cord and lower brain stem. On suspect diagnosis of seronegative NMOSD, he was treated with monoclonal antibody targeting CD20 B cells for five years without relapse. It took only 18 months without the treatment for the patient to relapse with both myelitis and bilateral ON, which confirmed the diagnosis [18,19]. During the first five years’ follow-up with OCT, the patient showed bilateral normal RNFL and GCIPL thickness. After the latest relapse, OCT showed significantly thinned RNFL and GCIPL. MRI showed an enhanced lesion in the chiasm which corresponded to a bitemporal VF defect; AQP4 and MOG IgG antibodies were still negative in both CSF and serum. This patient showed significant abnormalities in structure visualized by both MRI and OCT. Visual tests with perimetry and VEP predict permanent functional loss [27]. The sensitivity of MRI to detect ON is low (20–44%) and enhanced MRI lesions are specific for acute ON but can not often be caught [12,21]. AQP4 antibodies are highly specific in the diagnosis of NMOSD, which can delay seronegative NMOSD diagnosis and treatment. OCT exhibits bilateral axonal-neuron loss and provides useful markers to make a subclinical ON diagnosis and monitor clinical course.

Case 3 presents typical clinical manifestations of IIH with obesity, pain in the forehead, bilateral blurred vision and bilateral optic disc swelling in child-bearing age [28,29]. The patient lacked eye pain and had no dyschromatopsia challenging ON diagnosis, while CSF opening pressure via lumbar puncture was normal, speaking against IIH. Further development of VF defect, though the subjective dominant test, showed typical changes of ON course. MRI enhanced lesion in both optic nerves in the orbit and positive MOG IgG in the serum confirmed the diagnosis of NMOSD [17,30]. Symmetrically thickened RNFL corresponds to optic disc swelling. It took 6 weeks for RNFL to become atrophic and remained unchanged at three years’ follow-up. GCIPL was symmetric and normal at onset, but atrophic at week 6 and remained unchanged. RNFL is dynamically thickened during acute ON and IIH due to swollen optic nerves, which can be difficult to distinguish from each other with fundoscopy. It took more than 3 months to observe atrophy of the optic nerve after ON. Normalization of RNFL can be seen earlier in IIH course if treated appropriately, for example with acetazolamide [31,32]. RNFL atrophy occurs rarely in IIH. GCIPL is in general unchanged during IIH course unless severe untreated intracranial hypertension with visual loss is present. Such different OCT patterns can be useful for differential diagnosis between IIH and ON.

## 4. Conclusions

In conclusion, the absence of typical clinical ON manifestations can be challenging to make the diagnosis. VEP abnormality and MRI enhancement vary over time and often vanish after acute ON, but structural changes visualized by OCT remain after acute ON. With OCT, it is easy to detect the thickening of RNFL in acute ON, thinning of RNFL and GCIPL after acute ON. Moreover, OCT is a sensitive, objective and reliable measure in contrast to perimetry and color vision tests. Extending OCT’s role in diagnosis of atypical/subclinical or chronic ON is warranted.

## Figures and Tables

**Figure 1 jcm-12-01309-f001:**
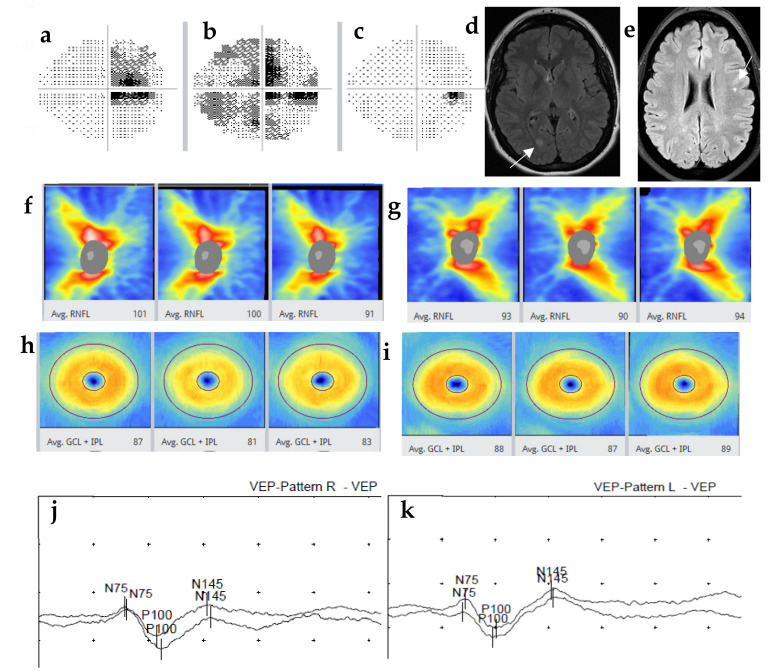
A 34-year-old woman with right side atypical optic neuritis (ON) at onset and who developed multiple sclerosis (MS) four years later. Status demonstrated with visual field (VF) test, optical coherence tomography (OCT) and visual evoked potentials (VEP). VF test showed superior quadrantanopia in the right eye at day 1 (**a**), central and peripheral VF defect at day 7 (**b**), but normalized at 2 months (**c**). MRI showed a T2 FLAIR lesion in the right occipital lobe at onset (**d**) and a juxtacortical lesion in the U fibers on the left side (**e**) 4 years later. OCT thickness map showed normal but asymmetric thickness of RNFL (**f**,**g**) and normal GCIPL (**h**,**i**) in both eyes at day 7 and week 8, but with thinning in RNFL (from 100 to 91 µm) and GCIPL (from 87 to 83 µm) in the right eye, and stable in the left eye at 5 months’ follow-up. VEP showed delayed latency of P100 in the right eye and normal in the left eye (right 111, left 101 ms) (**j**,**k**) at the onset which remained unchanged at 5 years’ follow-up.

**Figure 2 jcm-12-01309-f002:**
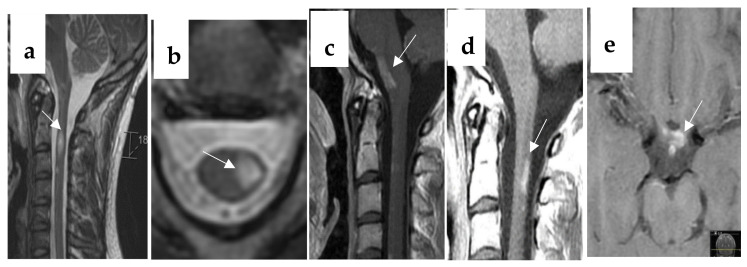
A 29-year-old man with seronegative neuromyelitis optica spectrum disorders (NMOSD) had bilateral subclinical optic neuritis (ON) demonstrated with MRI, OCT and VEP. Sagittal ((**a**), arrow) and axial ((**b**), arrow) T2 lesion lateral to the left at C2–3 at the onset; sagittal ((**c**), arrow) T1 lesion with enhancement in the ventral medulla oblongata at the second relapse; sagittal T1 enhanced lesions at C2–3 dorsal lateral to the left side ((**d**), arrow) and axial enhanced T1 lesion in the optic chiasm ((**e**), arrow) at the third relapse; normal thickness of RNFL (right 95 µm, left 91 µm) and GCIPL (right 79 µm, left 77 µm) (**f**); normal latency of P100 (right = left = 103 ms) (**g**) at 12 months without rituximab; thinned RNFL (right from 95 to 69 and left from 91 to 73 µm) and GCIPL (right from 79 to 56 µm, left from 77 to 63 µm) (**h**) and delayed latency of P100 (right 137 ms, left 135 ms) (**i**); bitemporal inferior quadrantanopia ((**j**) left, (**k**) right) at 18 months without rituximab.

**Figure 3 jcm-12-01309-f003:**
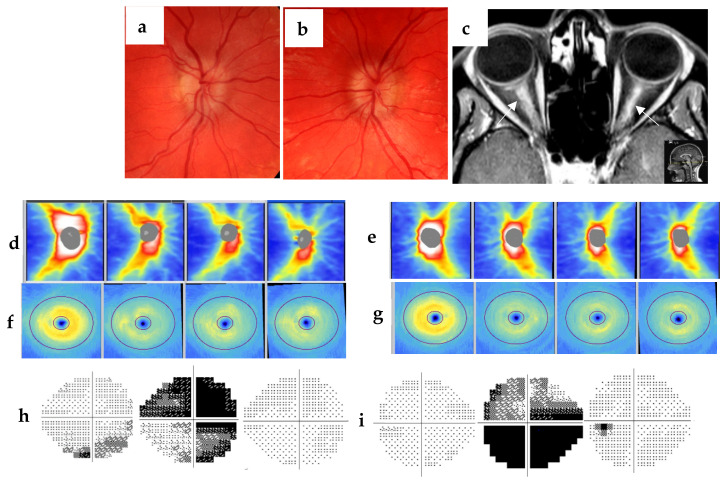
A 23-year-old obese female with positive myelin oligodendrocyte glycoprotein (MOG) antibody had bilateral subclinical ON demonstrated with MRI, OCT and VF. Fundoscopy showed papilledema in both right (**a**) and left eye (**b**). Orbital MRI visualized pathologic enhancement in the intraorbital optic nerves bilaterally (**c**). OCT thickness map showed significantly thickened RNFL in both right (152 µm) (**d**) and left (126 µm) (**e**) eyes at day 7, which was thinned at week 6 (right 100 µm, left 92 µm) and stable at follow-up of 3. 5 months and 3 years after onset. At day 7, GCIPL thickness was normal (right 74 µm, left 75 µm) (**f**,**g**), which was thinned at week 6 (right 66 µm, left 64 µm) and remained unchanged at 3. 5 months and 3 years. VF test (**h**,**i**) on day 7 demonstrated peripheral scotoma in the right (VFI 88%) and normal in the left eye (VFI 99%). VF was severely reduced in both eyes on day 14 (VFI right 26%, left 21%) but recovered at week 6 (VFI right 98%, left 94%).

## Data Availability

I confirm that my article contains a Data Availability Statement even if no data are available (list of sample statements) unless my article type does not require one (e.g., Editorials, Corrections, Book Reviews, etc.).

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
