# Peer review of "The Importance of Optical Coherence Tomography in the Diagnosis of Atypical or Subclinical Optic Neuritis: A Case Series Study"

_jcm, 2023, doi:10.3390/jcm12041309_

Round 1

Reviewer 1 Report

Interesting case presentations of optic neuritis in the contexts of CIS and NMOSD demonstrating the important diagnostic role of OCT for the diagnosis of optic neuritis in relation to other ophthalmological parameters, and MRI findings. Nevertheless, it should be noted that the diagnosis of the disease and the treatment were based on the antibodies found. The discussion and conclusions can be improved. 

Author Response

Responses to Reviewer 1.

We are very grateful for the reviewer's expert evaluations to make the improvement of our manuscript possible.

We have checked all references, and certain rearrangements have been done to make the references more relevant to the contents. We also added additional references corresponding to the modification of the manuscript. We used ”Track Changes” to highlight the changes.

Please find below our responses to the comments from the referee. The responses are point by point with corresponding changes in the manuscript.

  1. We have added additional information on the relationship between ON and MS in the Introduction with new references (Ref. 1, 9, 11).
  2. Additional information on OCT parameters is added with new references (Ref. 14,15) in the Introduction.
  3. According to the referee’s suggestion, the role of AQP4 antibody in the diagnosis of NMOSD has been mentioned in the Introduction. We have also modified the Conclusion.

Reviewer 2 Report

The manuscript encompasses descriptions of three cases of patients with optic neuritis. The text is well written and most of the important clinical details are provided (see below for some remarks). However, the novelty of the work is rather low, which is clearly shown by authors themselves in the Introduction. Subclinical involvement of optic tract in patients without history of optic neuritis was presented in earlier papers.

Regarding Case 2. The authors should comment on the possibility of the false negative results of anti-AQP4 and anti-MOG testing in the patient examined shortly after corticosteroid treatment. The methodology of immune test should be provided (was it a CBA test?). Moreover, the clinical description of this patients strongly suggest the need of reexamination of anti-AQP4 and anti-MOG antibodies. Were the tests repeated?

Regarding Case 3. The authors should refer to the actual literature and consider MOG associated disease (MOGAD) diagnosis in this case instead of NMO-ON. This would be in agreement with the classification suggested in the publication by Petzold et al. 2022, cited by the authors.

Author Response

We are very grateful for the reviewers’ expertise and evaluations to make the improvement of our manuscript possible.

We have checked all references, and certain rearrangement has been done to make the references more relevant to the contents. Additional five references (ref. 1, 9, 11, 14 and 15) corresponding to the modification of the manuscript have been added. We use ”Track Changes” to highlight the changes in the manuscript (please see the attachment).

Please find below our responses to the comments from the reviewers. The responses are point by point with corresponding changes in the manuscript.

1)   There is a lack of consensus on ON diagnostic criteria, which may increase misdiagnosis and delay therapy. We present 3 cases of atypical or subclinical ON to emphasize the important role of OCT as a paraclinical test in the diagnosis of ON in the lack of typical clinical features or neuroimaging findings, negative results in CSF and serum or normal eye fundus. In order to emphasize our focus, additional information and references (N0. 1, 9, 11, 14 and 15) have been added in the Introduction, Methods and Materials, Discussion and Conclusion. We hope that with these revisions, the novelty is improved.
2)   According to the comments of the Referee, we have added information about the methods and the time points of test for anti-AQP4 and anti-MOG IgG in Case 2. NMOSD antibodies including AQP1, 4 and MOG had been tested in both serum and CSF before the therapy with corticosteroids. Tests of NMOSD antibodies and CSF oligoclonal bands had been repeated 4 times during the disease course.
3)  We have added the diagnosis of MOGAD and ON classification proposed by Petzold et al in Case 3 with the corresponding reference according to the Reviewer’s comments.

Round 2

Reviewer 2 Report

The authors addressed properly all the indicated issues. In consequence the level of novelty can be also assessed as higher than in the previous review.